# Interplay between Children’s Electronic Media Use and Prosocial Behavior: The Chain Mediating Role of Parent–Child Closeness and Emotion Regulation

**DOI:** 10.3390/bs14060436

**Published:** 2024-05-23

**Authors:** Xiaocen Liu, Shuliang Geng, Donghui Dou

**Affiliations:** 1College of Preschool Education, Capital Normal University, Beijing 100048, China; 2223102004@cnu.edu.cn; 2School of Sociology and Psychology, Central University of Finance and Economics, Beijing 100081, China; psychaos@126.com

**Keywords:** electronic media, parent–child relationship, emotion regulation, prosocial behavior, social interaction, children

## Abstract

In the contemporary digital milieu, children’s pervasive engagement with electronic media is ubiquitous in their daily lives, presenting complex implications for their socialization. Prosocial behavior, a cornerstone of social interaction and child development, is intricately intertwined with these digital experiences. This relation gains further depth, considering the significant roles of parent–child relationships and emotion regulation in shaping children’s social trajectories. This study surveyed 701 families to examine the association between children’s electronic media use and prosocial behavior, specifically exploring the mediating roles of parent–child closeness and emotion regulation. Structural equation modeling was employed for the analysis. Children’s electronic media use negatively correlated with prosocial behavior, parent–child closeness, and emotion regulation. In contrast, a positive association emerged between parent–child closeness, emotion regulation, and prosocial behavior. Emotion regulation also correlated positively with prosocial behavior. Statistical analyses revealed that parent–child closeness and emotion regulation function as both individual and sequential mediators in the relation between electronic media use and prosocial behavior. The study’s analyses reveal that fostering children’s prosocial behavior in the digital era requires strong family ties, effective emotional management, and balanced digital exposure, which are pivotal for their comprehensive development.

## 1. Introduction

The post-pandemic era has witnessed a significant rise in electronic media use among children, altering their daily interactions and educational methodologies. The rapid spread of coronavirus disease 2019 (COVID-19) has exerted a profound impact on human life across the globe [1]. To mitigate the spread of the virus, educational institutions from kindergartens to secondary schools universally embraced online learning [2], thereby integrating electronic media deeply into children’s conventional living and learning paradigms. Research has highlighted a downturn in children’s physical and social activities during the pandemic, coupled with a noticeable increase in time allocated to electronic devices such as televisions, computers, and mobile phones [3]. Nagata et al. discovered that adolescents’ average daily screen time soared to 7.70 h during the early pandemic period, markedly higher than pre-pandemic figures, within a cohort of 5412 U.S. adolescents aged 12 and 13 [4]. Additionally, Carroll et al. found that, based on parent-reported data, 87% of Canadian children experienced a screen time surge [5]. This change in children’s electronic media consumption did not recede. Rather, it persisted beyond the reopening of schools [6], indicating a potential long-term shift in their media use patterns.

Electronic media use encompasses a spectrum of digital platforms, extending beyond traditional mediums like television and DVDs to include mobile phones, computers, tablets, and an array of interactive and streaming services [7]. This expansion reflects a broader societal pivot toward digital integration. The diverse implications of electronic media on children’s development have been extensively studied, yielding many outcomes across various domains. Feng et al. have noted potential benefits, such as enhancing spatial attention and reducing gender disparities in spatial cognition through action video games [8]. Conversely, findings by Zoromba et al. suggest that excessive media exposure may be associated with hyperactivity, anxiety, and learning difficulties in children [9]. Similarly, research by Raheem et al. has shown negative correlations between the duration of electronic media exposure and developmental aspects such as attention and language [10]. Given these varied findings, there is a pressing need for focused inquiry into electronic media use by children who have experienced the COVID-19 pandemic, to guide the development of informed media usage behaviors.

As digital technology entrenches itself in the fabric of childhood, a deeper understanding of electronic media use and its association with critical developmental outcomes like prosocial behavior becomes increasingly important. Prosocial behavior, characterized by voluntary actions intended to benefit others, such as care, help, consideration, comfort, and sharing, is a vital indicator of healthy social development in children [11]. Notably, positive social development, including prosocial behavior, is linked to enhanced interpersonal relationships, academic achievement, and cognitive abilities later in life [12,13]. Prosocial behaviors emerge early, around 1–2 years, and become more varied and frequent between ages 3 and 6, a critical period for social norm internalization [14]. The Social Context Model further emphasizes the significance of normative behaviors like prosocial actions for group acceptance and integration, underscoring their importance for social development [15]. This study delves into the complex association between electronic media use and the development of prosocial behavior in children, with particular attention to the roles of parent–child closeness and emotion regulation. In examining these relations, the research highlights the need for supporting prosocial development in children, particularly in an era where digital interactions are increasingly shaping their experiences and social dynamics.

### 1.1. Electronic Media Use and Prosocial Behavior in Children

The interplay between electronic media use and children’s prosocial behavior is multifaceted, drawing continued attention and scholarly debate. A segment of this research suggests a positive correlation. For example, Coyne et al. conducted a meta-analysis that uncovered a substantial relation between multidimensional prosocial media content and an increase in children’s prosocial behaviors [16]. Additionally, Stone et al. noted that the multiplayer collaborative nature of many video games could enhance cooperative skills, such as social interaction and information sharing [17]. Prot et al. similarly reported a positive association between prosocial media use and helpful behaviors in children, suggesting that certain types of electronic media use could support the development of prosocial tendencies [18]. Furthermore, Ostrov et al. found that among a sample of children of relatively high socio-economic status who were frequently exposed to educational programming like PBS Kids, there was an association between the amount of television viewed and concurrent prosocial behavior [19].

In contrast, other studies have offered a more nuanced view, suggesting potential negative outcomes associated with extensive electronic media use. Poulain et al. identified a link between excessive media use in children and reduced prosocial behavior [20]. Guo observed a negative correlation between media violence exposure and prosocial behaviors [21], resonating with earlier findings by Veraksa et al., who noted a relation between electronic media use, increased aggressive behavior, and decreased prosocial behavior [22]. Adding to this perspective, Lissak provided psychoneurological evidence indicating an inverse correlation between screen time and social skills [23]. Moreover, Christensen discussed how instant rewards in video games, enhancing gaming pleasure and triggering dopamine release, can lead to electronically mediated addictive behaviors and frontal lobe structure changes [24], impacting cognitive processes crucial for prosocial behaviors like empathy [25]. The General Learning Model (GLM) suggests that escalating electronic media use might lead to addiction-like relationships, potentially hindering social development [12,26]. Wiegman and van Schie highlighted a significant negative correlation between video game use and prosocial behavior [27]. Such evidence implies that the nature of media content and the extent of its use play significant roles in influencing children’s prosocial behavior.

Based on these findings, Hypothesis H1 is formulated in this study: There is a negative correlation between children’s electronic media use and prosocial behavior. It is postulated that increased engagement with electronic media may correspond to diminished real-world prosocial interactions among children.

### 1.2. Relations among Children’s Electronic Media Use, Parent–Child Closeness, and Prosocial Behavior

The construct of parent–child closeness, characterized by support, warmth, and a shared willingness for open communication, is a cornerstone of the parent–child dynamic [28,29]. Empirical evidence has illuminated the significant impact of children’s electronic media use on the quality of this relationship. Studies such as those by Zhu et al., Horita et al., and Ahmadian et al. have consistently shown that higher levels of problematic internet use among children are linked to poorer parent–child relationships [30,31,32]. The Displacement Hypothesis posits that excessive use of electronic media may supplant meaningful face-to-face interactions, eroding intimacy and satisfaction within familial relationships [33]. Similarly, Sampasa-Kanyinga et al. reported that intensive social media use can diminish parent–child communication and weaken the positive bonds within these relationships [34]. Adding to this, Nergiz et al. found that excessive screen time is positively associated with parental neglect, potentially leading to a decrease in parent–child closeness [35].

Conversely, a robust body of research underscores a positive association between parent–child closeness and children’s prosocial behavior. The Attachment Inner Work Model suggests that the quality of the parent–child bond lays the groundwork for the child’s interpersonal relationships and behaviors [36]. Xu et al. demonstrated that nurturing parent–child relationships is fundamental to preschoolers’ social interactions and significantly influences their subsequent social development [37]. Confirming this, Liu and Wang found a positive correlation between parent–child closeness and prosocial behaviors in a study encompassing 507 young children, with considerations for genetic factors [28]. Furthermore, research conducted in Ireland on 1,151 families illustrated that parent–child closeness significantly predicted prosocial behaviors in children, even when controlling for a spectrum of demographic variables [38]. This positive trend continues into adolescence, as indicated by Padilla-Walker et al., who observed that warmth and connection in the parent–child relationship are positively related to adolescents’ prosocial behavior [39]. Similarly, Ferreira et al. emphasized the link between the quality of early caregiver relationships and children’s prosocial behavior [11].

Therefore, this study introduces Hypothesis H2, which posits that parent–child closeness acts as a mediating factor in the relation between children’s electronic media use and their prosocial behavior. Specifically, it is hypothesized that increased electronic media use among children may be associated with decreased parent–child closeness, which may reduce prosocial behaviors. This mediation hypothesis suggests that parent–child closeness is a critical mediator that could explain the link between children’s electronic media use and prosocial conduct.

### 1.3. Interconnections among Children’s Electronic Media Use, Emotion Regulation, and Prosocial Behavior

Emotion regulation is pivotal for maintaining both physical and mental well-being. It involves modulating one’s emotional responses adaptively through various strategies to alter the intensity and duration of emotions [7,40]. The nexus between electronic media use and emotion regulation has increasingly become scrutinized with the rise in media accessibility. Lobel et al. found in their longitudinal study of 7 to 11-year-olds that children’s gaming on electronic devices correlated with heightened emotional difficulties [41]. Günaydin et al. and Özer et al. corroborated this by identifying a significant link between problematic internet use and challenges in emotion regulation [42,43]. Moreover, Oflu et al. emphasized that excessive screen time was associated with the negative development of emotional regulation in early childhood [44]. Echoing these findings, Hidayatullah et al. reported that internet-addicted individuals exhibited significant difficulties with emotion regulation [45]. This trend persisted across various demographic groups, including differences in gender, age, and educational level [46]. 

The relation between children’s emotion regulation and prosocial behavior has been extensively studied, revealing a consistent pattern of associations. Dunfield noted that individual differences in emotion regulation abilities significantly influence children’s propensity for prosocial actions [47]. This finding aligns with Benita et al., who identified effortful control, an essential aspect of emotion regulation, as facilitating empathy and prosocial behavior in social contexts [48]. Fabes et al. further elucidated that positive emotion regulation aids children in being attentive to and empathetic towards others’ suffering, thereby enhancing their prosocial inclinations [49]. Additionally, effective emotion regulation enables children to better manage their responses in distressing scenarios, making them more likely to offer help and support [50]. Similarly, Song et al. found in their study with 4–8-year-old children that the ability to regulate grief was positively associated with mother-reported prosocial behavior [51].

Based on these observations, Hypothesis H3 is proposed: children’s use of electronic media is negatively associated with their capacity for emotion regulation, which in turn is positively associated with prosocial behavior. The hypothesis further suggests that emotion regulation may mediate the relation between electronic media use and prosocial behavior in children.

### 1.4. Associative Patterns among Children’s Electronic Media Use, Parent–Child Closeness, Emotion Regulation, and Prosocial Behavior

Research has shed light on the relation between parent–child closeness and emotion regulation in children. Dysfunctional family environments or negative parent–child relationships have been shown to disrupt emotion regulation, leading to increased insecurity in individuals [52,53]. In contrast, children with secure attachment, which fosters improved emotion regulation, are likely to exhibit enhanced empathic responding and prosocial behavior [54]. Strong parent–child bonds, characterized by quality interactions, are essential for developing effective emotion regulation. These interactions often involve emotional synchronization, positively contributing to children’s emotional growth [55,56]. Zhao et al. emphasized how early mother–child interactions develop synchrony across physiological, neurological, and behavioral aspects, influencing children’s emotion regulation abilities [57].

Building on Bowlby’s Attachment Theory, Ainsworth introduced an emotional dimension to the parent–child attachment relationship, emphasizing its profound influence on children’s emotion regulation and social adjustment [58]. This early childhood relationship is crucial in forming children’s self-perception and understanding of others. Bronfenbrenner and Morris further demonstrated that positive parent–child relationships are instrumental in nurturing children’s socio-emotional regulation, influencing their ability to form healthy interpersonal relationships [59]. On the other hand, negative parent–child interactions can lead to emotional dysregulation and foster negative perceptions in children about interpersonal relationships, potentially leading to a reduction in prosocial behaviors such as cooperation and helping [28,60].

Considering the insights gained from the preceding review, a chain mediation hypothesis is proposed. This hypothesis, Hypothesis H4, suggests that parent–child closeness and emotion regulation play sequential mediating roles in the relation between children’s electronic media use and prosocial behaviors. Specifically, this hypothesis considers the possibility of an association where higher levels of children’s electronic media use might be related to lower parent–child closeness. In turn, lower parent–child closeness might be associated with less-effective emotion regulation, which could correlate with decreased prosocial behaviors among children.

### 1.5. The Present Study

In the burgeoning field of child development research, the comprehensive effects of electronic media have become increasingly prominent. Despite extensive exploration, the majority of existing studies predominantly focus on the adverse developmental outcomes associated with children’s electronic media use, such as problematic behaviors. However, there is a noticeable dearth of research delving into the positive aspects of development, particularly prosocial behavior, concerning children’s electronic media engagement. This gap is significant considering the pivotal role of early childhood in cultivating social and emotional competencies, often nurtured through family interactions. Our study seeks to fill this gap by investigating the specific impact of children’s media use within the family context, exploring its relation with parent–child closeness, emotion regulation, and the development of prosocial behaviors.

Moreover, the previous literature has frequently concentrated on the impact of parental media habits on the parent–child relationship. In contrast, our study focuses on how children’s interactions with media contribute to this dynamic, providing a more comprehensive understanding of the family media environment. We frame our exploration using Knafo and Plomin’s Individual–Environment Interaction theory [61]. This theory suggests that children’s development is influenced by a blend of individual characteristics, such as media use and emotion regulation, and environmental factors, like parent–child closeness. Additionally, we integrate the concept of the Bi-Directionality of Parent–Child Relationships theory, highlighting the reciprocal nature of these interactions. This perspective emphasizes the co-contributory roles of both parents and children in shaping their relationship dynamics. Through our research, we aim to illuminate how children’s electronic media use, within the context of their family environment, influences and is influenced by their emotional and social development.

Our study addresses a critical knowledge gap by exploring the positive developmental outcomes associated with electronic media use in children, particularly focusing on prosocial behavior. Employing chain mediation analysis and grounded in established theories, our research delves into the associations and interrelationships between children’s use of electronic media, their prosocial behaviors, and the roles of parent–child closeness and emotion regulation as mediators. This study aims to provide a deeper understanding of these associations within the digital age context. The chain mediation model, depicted in Figure 1, is designed to elucidate these interconnections, offering insights into the complex dynamics of electronic media’s role in children’s social and emotional development.

## 2. Materials and Methods

### 2.1. Participants and Procedure

For this study, we collaborated with teachers from kindergartens and primary schools across northern China to facilitate the distribution of questionnaires. From December 2022 to February 2023, these questionnaires were administered to parents via an online survey platform. The selection criteria included ensuring that the children and their parents were free from any psychiatric or neurological conditions, as reported by the teachers. Before participants filled out the questionnaire, we provided them with a comprehensive briefing. This briefing included detailed definitions and the scope of electronic media use to ensure a common understanding of the terms used in the study. Informed consent was obtained from all participants, affirming their voluntary participation and awareness of the study’s objectives and procedures.

We conducted rigorous screening to ensure data integrity and the relevance of the responses. The initial phase of our study involved the distribution of 796 questionnaires. Questionnaires were excluded if they contained inconsistencies, such as affirmative responses to the lie detector statement “My child never blinks”, which served to identify non-serious responses. We also excluded questionnaires with contradictory responses (e.g., claiming “The child has never been exposed to electronic media” but later reporting media usage), incomplete entries, or those not filled out by the child’s primary caregivers. After applying these criteria, we obtained 701 valid questionnaires for our final analysis.

To validate the robustness of our statistical conclusions, we conducted a post hoc power analysis using G*Power 3.1.9.7 software. We aimed for an effect size of at least a small magnitude (*f*^2^ = 0.02) with an alpha level (α) set at 0.05. This analysis showed that our sample of 701 participants provides a power value of 0.84, surpassing the commonly accepted threshold of 0.80 [62]. The power analysis result confirmed that the sample size of 701 is sufficiently large to detect small effect sizes, thus ensuring adequate statistical power for the statistical tests employed. Moreover, according to guidelines by Bentler et al., the sample size for structural equation modeling should ideally range from 5 to 10 times the number of estimable parameters [63]. With 47 estimable parameters in our model, the recommended sample size would be between 235 and 470. Thus, our sample size meets and comfortably exceeds these recommendations, enhancing our findings’ reliability and validity and ensuring the structural equation model’s robustness.

The final participant pool comprised 190 fathers (27.1%), with an average age of 34.09 years (*SD* = 5.11), and 511 mothers (72.9%), with an average age of 32.33 years (*SD* = 4.63). The predominance of mothers in the sample can be attributed to the traditional gender roles prevalent within Chinese families. Historically, societal expectations have designated men as the primary breadwinners, involved chiefly in the public sphere. At the same time, women are typically seen as the principal caregivers, responsible for domestic roles, including child-rearing. This cultural norm has naturally led mothers to assume a more substantial role in managing children’s daily needs and activities, resulting in their deeper involvement and understanding of the children’s routines and challenges [64]. However, contemporary societal shifts and changes in family dynamics are gradually altering these traditional roles. An increasing number of fathers are now actively engaging in parenting, spending more time with their children, and sharing the responsibilities traditionally held by mothers. It is important to note that all participants in this study were primary caregivers who resided with the children, ensuring they were well-informed about their daily lives and capable of providing reliable responses.

Further enriching the demographic profile, the educational qualifications of the participants vary significantly, offering a broad perspective on parental influence. Concerning the educational qualifications of the children’s mothers, 2.8% attained junior high school level or below, 8.5% completed high school or junior college, 78.4% held a college or bachelor’s degree, and 10.3% were postgraduate students. Similarly, for the fathers, 2.7% had a junior high school education or below, 9.4% had completed high school or junior college, 70.4% had a college or undergraduate degree, and 17.5% had postgraduate qualifications. Additionally, family structure within the sample was split between 54.6% nuclear families and 45.4% extended families, illustrating the diversity of family setups influencing child development.

The children involved in this study had a mean age of 5.20 years (*SD* = 1.87), consisting of 339 boys (48.4%) and 362 girls (51.6%). The sample included 547 singletons (78.0%) and 154 children with siblings (22.0%). Concerning their educational settings, 6.7% of the children were not enrolled in any educational institution, 81.7% attended kindergartens, and 11.6% were enrolled in primary schools. To ensure the reliability of our findings, we only included children and parents with no psychiatric or neurological conditions, as verified by the teachers at the time of recruitment. This careful selection was crucial to ensure that the observed behaviors and interactions were typical of the general child population, unaffected by underlying health issues.

### 2.2. Measures

#### 2.2.1. Electronic Media Use

Our recent study assessed children’s interaction with electronic media using an adjusted version of the Children’s Electronic Media Use Questionnaire. Building upon foundational work by Huang et al. [65], further elaborated on by Geng et al. [66], this instrument consists of 14 items designed to evaluate various aspects of children’s media usage. For example, it includes questions such as “My child throws tantrums because I limit her/his time to use electronic media”, highlighting emotional reactions to restricted media access. Responses to these items were recorded on a 5-point scale to quantify the extent of media engagement. The questionnaire’s reliability in our study was evident, with a Cronbach’s α value of 0.93, indicating high internal consistency.

#### 2.2.2. Parent–Child Closeness

In assessing the parent–child intimacy, our study utilized the closeness subscale from the Child–Parent Relationship Scale, originally conceptualized by Pianta [29] and later adapted by Xu and Wang [67]. The closeness aspect of this scale is particularly insightful, encompassing ten items that gauge the warmth and affection in the parent–child relationship (e.g., “My child values his/her relationship with me”). These items are rated on a 5-point scale, where higher scores indicate a stronger, more positive bond between parent and child. In our study, the closeness subscale exhibited strong internal consistency, as evidenced by a Cronbach’s α value of 0.85.

#### 2.2.3. Emotion Regulation

In this study, we utilized the Emotion Regulation Checklist, originally developed by Shields and Cicchetti [68] and later revised by Xing et al. [69], with a particular emphasis on its emotion regulation subscale for assessing this facet of children’s abilities. The subscale comprises eight items that measure children’s adaptability and response to emotional challenges. An example is the item “Can recover quickly from upset or distress (for example, doesn’t pout or remain sullen, anxious, or sad after emotionally distressing events)”. These items were rated on a 4-point scale, with higher scores indicating more effective emotion regulation. In our research, the emotion regulation subscale demonstrated sound internal consistency, as reflected by Cronbach’s α value of 0.73.

#### 2.2.4. Prosocial Behavior

The Strengths and Difficulties Questionnaire (SDQ), known for its robust reliability in assessing mental health across various child age groups, was employed to evaluate children’s prosocial behavior. Formulated initially by Goodman [70] and adapted for broader applicability by Aarø et al. [71], this questionnaire includes 21 items spanning three key dimensions: prosocial behavior, externalizing problems, and internalizing problems. For this research, we selectively focused on seven items from the prosocial behavior dimension. For example, my child shares readily with other children (treats, toys, pencils etc.). These items were rated on a 3-point scale, where 0 signifies “not true” and 2 denotes “certainly true”. A higher tally on this subscale suggests a more frequent occurrence of prosocial behaviors like sharing. The Cronbach’s α value for this specific subscale in our research was 0.80, indicating a satisfactory level of internal consistency.

### 2.3. Data Analysis

In this study, SPSS software (version 23.0) served as the primary tool for the initial stages of data handling, encompassing input, organization, and preliminary assessments such as checking for common method bias, evaluating data normality, and verifying scale reliability. We then conducted descriptive and correlational analyses to ascertain the scores of each variable and explore their interrelationships. Subsequently, to build upon this initial data exploration, structural equation modeling (SEM) was employed using AMOS (version 26.0). This advanced technique detailed the associations among children’s electronic media use, parent–child closeness, emotion regulation, and prosocial behavior.

## 3. Results

### 3.1. Common Method Bias and The Normality of the Data

To address potential common method bias due to the self-reported nature of our data, we implemented measures such as anonymous response collection and the inclusion of reverse-scored items. In further evaluating common method bias, we applied Harman’s single-factor test, as Podsakoff et al. suggested [72]. The exploratory factor analysis results revealed six factors with eigenvalues exceeding one. Notably, the variance attributed to the most substantial common factor was 25.48%, falling below the threshold of 40%. This finding suggests that common method bias did not significantly influence our study’s results. Regarding data normality, we utilized Q–Q plots for assessment. The alignment of most data points close to the diagonal line in these plots substantiates our research’s assumption of data normality.

### 3.2. Descriptive Statistics and Correlation Analysis

This section provides a comprehensive overview of the demographic data and distribution characteristics of our study sample, detailing the socio-economic, familial, and individual attributes along with their associations with the study’s outcomes. To accurately categorize families and analyze their potential influence on child development, we calculated Family Socio-Economic Status (SES) using Reng’s formula [73]. This calculation incorporates the highest educational level of the parents, their occupation, and monthly family income as key indicators: SES = β_1_ × Z _higher education level_ + (β_2_ × Z _higher occupation_ + β_3_ × Z _monthly family income_)/εf.

Following this foundational classification, Table 1 provides a detailed breakdown of the children’s and families’ sociodemographic characteristics alongside the children’s scores for electronic media use and prosocial behavior. Notable findings include significant differences in prosocial behavior scores between genders, with boys scoring lower than girls (*p* < 0.001). Children under three years of age displayed significantly lower prosocial behavior scores compared to those aged 3–6 years and older than 6 years, with all comparisons yielding *p* < 0.001. However, no significant differences were observed between other sociodemographic characteristics and scores for electronic media use and prosocial behavior.

Building upon the demographic and socio-economic profiles outlined, we delve deeper into the data to explore various interactions and correlations, as detailed in Table 2. Notably, a negative correlation existed between children’s electronic media use and key variables such as parent–child closeness, emotion regulation, and prosocial behavior. In contrast, parent–child closeness positively correlated with emotion regulation and prosocial behavior in children. Additionally, children’s emotion regulation was positively correlated with their prosocial behavior. Furthermore, children’s age positively correlated with electronic media use and prosocial behavior. Intriguingly, the correlation between children’s gender and prosocial behavior was positive, suggesting that girls exhibited more prosocial behavior than boys. Given these insights, children’s age and gender, informed by the results above, were subsequently included as control variables in the model.

### 3.3. Chain Mediation Model Test

To analyze the structural equation model, AMOS 26.0 was employed to validate the mediating roles of parent–child closeness and emotion regulation. We implemented a bias-corrected confidence interval Bootstrap test for examining the model paths. This test involved 5000 resamples and set 95% confidence intervals, ensuring robustness in our findings. In line with the guidelines suggested by Little et al. [74] and Rogers and Schmitt [75] for item parceling in structural modeling, we adopted specific approaches for different scales. For the Electronic Media Use questionnaire, which contains multiple subscales, we used isolated parceling to combine these subscales into a single indicator. Conversely, a factorial algorithm approach was applied for scales with a single dimension, such as the parent–child closeness and emotion regulation subscales. Given the limited number of items in the prosocial behavior subscale, we opted not to employ the parceling strategy for this measure.

The initial stage of our mediation analysis was to investigate the relation between children’s usage of electronic media and their display of prosocial behavior, considering age and gender as potential influencing factors. The model fit, as measured by several indices, showed an appropriate level: χ^2^/df = 1.85, RMSEA = 0.03, NFI = 0.97, GFI = 0.98, IFI = 0.98, RFI = 0.96, CFI = 0.98, and TLI = 0.98. A statistically significant negative link was found between children’s electronic media usage and prosocial behavior (β = −0.19, *p* < 0.001, 95% CI [−0.27, −0.10]), implying that increased electronic media use tends to correspond with reduced prosocial behavior.

In further exploring the model, we included parent−child closeness and emotion regulation to understand their roles as mediators. The revised model also demonstrated a satisfactory fit (χ^2^/df = 2.35, RMSEA = 0.04, NFI = 0.94, GFI = 0.95, IFI = 0.97, RFI = 0.93, CFI = 0.97, and TLI = 0.96). Interestingly, the direct association between children’s electronic media use and prosocial behavior became non-significant (β = 0.04, *p* = 0.290, and 95% CI [−0.03, 0.12]) upon introducing parent−child closeness and emotion regulation into the model. However, as depicted in Figure 2, negative paths emerged from electronic media use to both parent−child closeness (β = −0.15, *p* < 0.001, and 95% CI [−0.23, −0.07]) and emotion regulation (β = −0.19, *p* < 0.001, and 95% CI [−0.26, −0.12]). Positive and significant pathways were observed from parent–child closeness to emotion regulation (β = 0.74, *p* < 0.001, and 95% CI [0.68, 0.79]) and prosocial behavior (β = 0.19, *p* < 0.05, and 95% CI [0.03, 0.32]), as well as from emotion regulation to prosocial behavior (β = 0.66, *p* < 0.001, and 95% CI [0.52, 0.82]). These findings indicate a complete mediation effect, suggesting the pivotal role of parent−child closeness and emotion regulation in the link between electronic media use and prosocial behavior.

To substantiate these mediating roles, we employed Bootstrap analysis for a more accurate estimation of confidence intervals (see Table 3). The mediating effect is significant if the 95% confidence interval does not include zero. The results showed a significant mediated effect from children’s electronic media use to prosocial behavior via parent–child closeness (standardized effect = −0.0282 and 95% CI [−0.06, −0.01]) and via emotion regulation (standardized effect = −0.1249 and 95% CI [−0.19, −0.07]). Moreover, the chain-mediated pathway—from electronic media use through parent–child closeness and emotion regulation to prosocial behavior—was also significant (standardized effect = −0.0744 and 95% CI [−0.12, −0.03]). This outcome illustrates the significance of the chain-mediated effect, suggesting a cascading influence: higher electronic media use in children is associated with reduced parent–child closeness and increased emotional challenges, adversely affecting prosocial behavior.

## 4. Discussion

This study aimed to validate the association between children’s electronic media use and prosocial behavior and explore the mediating role of parent–child closeness and emotion regulation in this connection. Our findings support the complexity of this interplay, revealing that electronic media use is intricately linked with prosocial behavior through both direct and mediated pathways involving parent–child closeness and emotion regulation.

### 4.1. Children’s Electronic Media Use and Prosocial Behavior

Consistent with the first hypothesis (H1), our analysis found a negative association between children’s electronic media use and prosocial behavior. Structural equation modeling unveiled that prosocial behavior tends to diminish as electronic media use escalates. This phenomenon may be attributed to the pervasive nature of electronic media, which, as documented by Ding et al., is becoming a significant health concern globally, increasingly affecting younger populations [76]. The World Health Organization, in the 11th edition of the International Classification of Diseases (ICD) published in 2018, has recognized video game addiction as a mental health disorder. Children, with their developmental cognitive and neural mechanisms, are particularly vulnerable to the adverse impacts of electronic media [23,76]. Excessive engagement with electronic media may disrupt crucial real-life interactions that foster the development of communication skills and prosocial behaviors such as empathy, sharing, and cooperation [22,77]. The displacement of face-to-face interaction time is particularly concerning, as it is through these direct social engagements that children learn and practice prosocial behaviors [78]. Such interpersonal experiences, often cultivated through real-life experiences and direct social engagement, may not be adequately stimulated in digitally mediated environments. Furthermore, the immersive qualities of electronic media can engender escapism, potentially diminishing children’s willingness to engage with real-world challenges and respond with prosocial behaviors [79].

### 4.2. Mediation of Parent–Child Closeness

Our investigation, corresponding to the second hypothesis (H2), indicated that parent–child closeness appears to mediate the relation between children’s electronic media use and prosocial behavior. The data suggest a pattern where elevated use of electronic media aligns with reduced levels of parent–child closeness, which is concurrently associated with lower indications of prosocial behavior in children. Insights into the association between children’s electronic media use and the quality of parent–child closeness emerge from examining the Displacement Hypothesis. This theory, as outlined by Hong et al., suggests that time spent engaging in electronic media might encroach upon and diminish valuable real-life interactions, leading to reduced intimacy in interpersonal relationships [33]. Supporting this theory, research by Nergiz et al. indicates that excessive access to electronic media could exacerbate parental neglect and undermine the development of parent–child closeness [35]. Furthermore, high media engagement levels might conflict with parental expectations for their children’s active involvement in real-life activities, leading to an “expectation deviation” that can negatively impact the parent–child bond [66]. Additionally, Beyens and Beullens noted the negative impact of extensive tablet computer use on developing a positive parent–child relationship [80].

The linkage between parent–child closeness and prosocial behavior can be understood through various psychological frameworks. The Emotional Security Hypothesis suggests that secure parent–child relationships are fundamental to children’s emotional well-being and social adaptation, while insecure relationships may contribute to difficulties in social adjustment [81]. This view is supported by the Attachment Inner Work Model, which posits parent–child closeness as foundational for an individual’s interpersonal relationships and behaviors [36]. Consistent with these perspectives, positive parent–child interactions have been shown to significantly predict children’s prosocial behavior, as seen in the findings of Liu and Wang [28]. Moreover, Attachment Theory provides a compelling framework for understanding the impact of parent–child relationships on children’s social development. Secure attachments, characterized by warmth and responsiveness, are essential for children’s development of empathy and cooperative behaviors, which are key components of prosocial conduct [82]. Conversely, insecure attachments may hinder these competencies. Supporting these findings, Katsantonis and Mclellan have demonstrated that high-quality parent–child relationships foster prosocial behaviors in children, underscoring the importance of positive family interactions in developing social competencies [83].

### 4.3. Mediation of Children’s Emotion Regulation

Hypothesis 3 (H3) of our study posited that emotion regulation may mediate the relation between children’s electronic media use and prosocial behavior. Confirmatory results from structural equation modeling indicated a notable association, suggesting that as children’s electronic media use increases, their capacity for emotion regulation may be compromised, which is associated with a lower propensity for prosocial behavior. This finding contributes to understanding how electronic media, which often require rapid and reactive attention shifts, may influence children’s emotional development. The cognitive load imposed by many electronic media tasks may tax children’s developing neural systems, leaving less capacity for the reflective control necessary for emotion regulation [84,85,86]. These findings align with the work of Lobel et al., which associated extensive gameplay on electronic devices with increased emotional problems among children [41]. von der Heiden et al. also reported that problematic video game use was moderately negatively associated with emotional and psychological well-being, further highlighting the potential impact of electronic media on children’s emotional competencies [87].

Beyond psychological explanations, recent research has revealed the physiological underpinnings that strengthen the association between emotion regulation and prosocial behavior in children. Eisenberg accentuated the role of cardiac vagal tone—a physiological marker of emotion regulation—in nurturing prosocial behavior, suggesting that a calm and well-regulated emotional state is likely instrumental in promoting prosocial actions [88]. In support of this notion, Song et al. provided complementary evidence, showing that children who excel in emotion regulation tend to exhibit more prosocial behaviors [51]. Likewise, Elhusseini et al. found that children skilled in managing their emotions demonstrate a heightened ability to attend to others’ needs and show an increased propensity for helping behaviors in difficult situations [50].

### 4.4. Sequential Mediation of Parent–Child Closeness and Children’s Emotion Regulation

Our investigation substantiates Hypothesis 4 (H4), suggesting that parent–child closeness and emotion regulation sequentially mediate the relation between children’s electronic media use and prosocial behaviors. Specifically, this hypothesis considers a pathway wherein elevated levels of electronic media engagement in children may correlate with diminished parent–child intimacy. This observed decrease in closeness is associated with a corresponding decrease in children’s emotion regulation capabilities, which may be related to a reduction in prosocial behaviors. The Parent–Child Emotion Regulation Dynamics model, developed by Morris and colleagues, provides a framework for understanding the association between parent–child relationships and children’s emotion regulation [89]. It emphasizes the dynamic, reciprocal processes of emotion regulation within the parent–child interaction, where both parties influence each other’s emotional states. This model is enhanced by neurobiological research, such as that by Ratliff et al., which includes cross-brain associations in dyadic emotion regulation during social-emotional interactions, highlighting the role of coordinated brain responses in shaping children’s emotion regulation [90]. Supporting this, studies like Liu et al.’s associate negative parent–child interactions with negative emotion regulation outcomes in children [91].

In the nuanced interplay between electronic media use and children’s prosocial behavior, the chain mediation effect of parent–child closeness and emotion regulation emerges as a pivotal factor. Geng et al. highlight how children’s intensified use of electronic media may clash with parental expectations, potentially diminishing parent–child intimacy and increasing the likelihood of conflict [66]. This disruption in closeness is a critical precursor to the observed impairments in emotion regulation as outlined by Cummings and Davies [52] and Fabes et al. [53], which, in turn, can lead to decreased prosocial behaviors among children. The findings of Criss et al. reinforce this cascade by delineating the strong link between the quality of parent–child relationships and children’s emotion regulation and subsequent behavior patterns [92]. Secure attachments, foundational for the development of effortful control as described by Gross et al., enable children to engage in other-oriented behavior, a cornerstone of prosocial action [93]. By eroding these attachments, excessive electronic media use may decrease positive parent–child interactions, reducing closeness and fostering emotional dysregulation. This sequence—beginning with electronic media use and flowing through diminished parent–child closeness and impaired emotion regulation—culminates in reduced prosocial behavior, outlining a clear chain-mediating pathway.

### 4.5. Limitations and Future Research Directions

In the present study, we explored how children’s electronic media use might inversely relate to their prosocial behaviors and how this relation is potentially mediated by parent–child closeness and the children’s ability to regulate emotions. These connections offer insights into the multifaceted influences of digital media consumption on the healthy development of children. Nevertheless, acknowledging the limitations enhances the robustness of the research.

Turning to the constraints of our study, we acknowledge that our analysis did not differentiate between various types of electronic media content such as video games, internet usage, or TV programs. While programs like “Baby Einstein” and “Sesame Street” are intentionally designed for children, providing educational content and themes suitable for their developmental stages, our study did not precisely control for these distinctions. Future studies should incorporate a comprehensive content analysis to deepen our understanding of how media influences children’s social and emotional well-being. Such an analysis would examine how various types of media—educational, entertainment, and social networking—differentially affect children’s development. By exploring the specific intentions behind media design, such as whether they aim to enhance learning or foster social skills, researchers can better assess how these influences contribute to or detract from children’s prosocial behaviors.

Beyond the content analysis, the methodological design of our research also warrants further discussion. Our study’s methodology is rooted in cross-sectional analysis, which provides a snapshot of the complex relations between children’s electronic media use and prosocial behavior. While useful, this approach is limited in its ability to ascertain the directionality or causality of these relationships. The intricate dynamics of how children’s media consumption patterns might be shaped by, and potentially shape, the quality of parent–child interactions call for a deeper, temporal examination. In light of this, adopting a longitudinal research design would be highly valuable, enabling observation of the long-term dynamics between children’s electronic media use and prosocial development and providing the evolution of these relations over time.

Additionally, while considering the data’s validity, the nature of its collection necessitates scrutiny. Parental self-reports, which served as the primary data source for this study, are susceptible to biases such as social desirability, where respondents may answer in a manner they perceive to be more favorable rather than reflecting their actual behaviors or feelings. Such biases can particularly distort insights into sensitive areas such as parent–child relationships. Thus, future research should adopt a multifaceted data collection strategy to mitigate these limitations and enhance the reliability of the findings. An improved research methodology would incorporate objective measures like digital usage logs and behavioral observations to complement self-reported data. Additionally, collecting data from multiple informants, including teachers and healthcare providers, could broaden the perspective and enrich the data quality. Furthermore, directly engaging with children alongside their parents would offer deeper insights into family dynamics and their impact on children’s prosocial behaviors and interactions with media. This comprehensive approach would provide a more holistic understanding of the factors influencing children’s development in a digital world.

In addition to the methodological enhancements, the development of children’s prosocial behavior is influenced by multiple factors, including temperament, which plays a crucial role. While the emotion regulation examined in this study partially reflects temperament, it is clear that a more detailed control of variables such as children’s temperament types is necessary for future studies to provide a more comprehensive understanding of prosocial development in the digital media era. Additionally, replicating this study with parents of older children would be insightful, particularly by comparing groups with preschoolers, school-aged children, and adolescents. This approach would allow us to explore how prosocial behavior evolves with age and how different stages of childhood might interact differently with media use.

### 4.6. Practical Implications

Drawing on the findings of our study, the practical implications for children’s development in the context of digital media are multifaceted. These implications, integrating the research outcomes, provide a framework for informed interventions and strategies to support children’s healthy development in the digital age.

Parent–child closeness is a critical factor in mediating the impact of electronic media use on children’s prosocial behavior. This finding underscores the importance of fostering strong parent–child relationships. Parents can be encouraged to engage more actively with their children’s media use, supervising and participating in their digital activities. This active involvement can provide opportunities for bonding, understanding each other’s perspectives, and guiding children toward positive digital practices.

Emotion regulation emerged as a crucial mediator between electronic media use and prosocial behavior. Educational interventions can be designed in schools and communities to enhance children’s emotional awareness and management skills. These programs can be integrated into the curriculum or offered as extracurricular activities, focusing on teaching children effective strategies for understanding and regulating their emotions, which are necessary for developing prosocial behaviors.

Considering the complex interplay between electronic media use and children’s development, promoting a balanced approach to digital consumption is essential. Parents and educators can collaborate to set reasonable limits on screen time while encouraging children to engage in various activities that support their social and emotional development. This balanced approach should include outdoor activities, reading, and face-to-face social interactions, pivotal for holistic growth.

In summary, the practical implications of our research highlight the need for comprehensive strategies involving parents and educators to support children’s development in the digital age. By enhancing parent–child closeness, improving emotion regulation skills, and promoting a balanced media diet, we can guide children toward positive developmental outcomes in the era of digital media.

## 5. Conclusions

Our study found a notable association between children’s electronic media use and prosocial behavior, underscoring the complex dynamics of digital media in the context of child development. The research not only highlighted the individual mediating roles of parent–child closeness and emotion regulation but also importantly confirmed their interconnected, sequential mediation effect in this relation. The results suggest that extensive involvement in electronic media might be linked to decreased parent–child closeness and could be associated with challenges in emotion regulation, which in turn appears to correlate with reduced prosocial behaviors in children.

## Figures and Tables

**Figure 1 behavsci-14-00436-f001:**
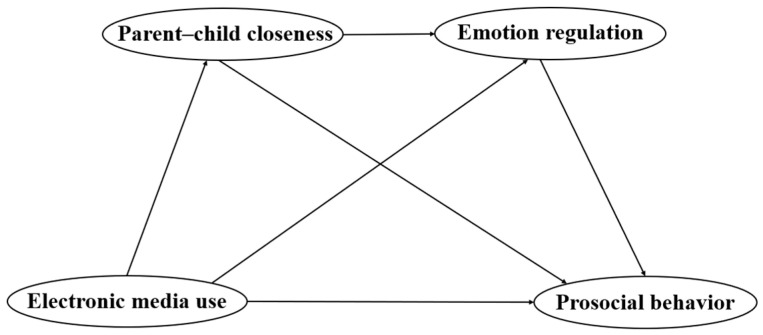
Hypothetical chain mediation model.

**Figure 2 behavsci-14-00436-f002:**
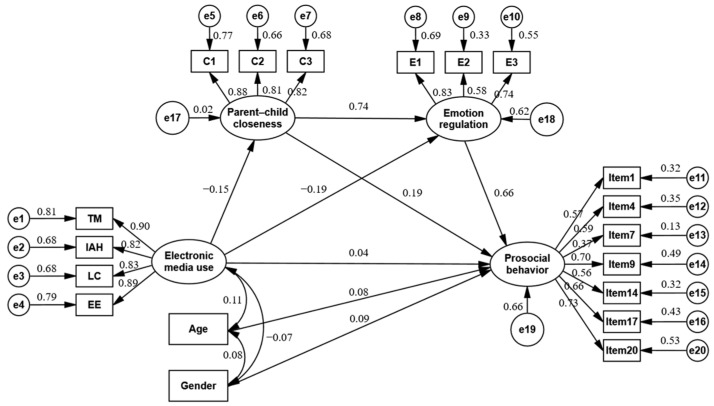
Standardized path diagram of relations.

**Table 1 behavsci-14-00436-t001:** Associations between sociodemographic features and scores for children’s electronic media use and prosocial behavior.

	*n* (%)	Electronic Media Use Score				*p*	Prosocial Behavior Score				*p*
		*Median*	*Mean*	*SD*	*IQR*		*Median*	*Mean*	*SD*	*IQR*	
**F** **amily structure**											
Nuclear families	383 (54.64%)	2.29	2.33	0.71	1.14	0.370	1.43	1.37	0.41	0.71	0.304
Extended families	318 (45.36%)	2.36	2.38	0.78	1.14		1.43	1.40	0.40	0.57	
**Family socio-economic status (SES)**											
Low SES	224 (32.0%)	2.43	2.43	0.74	1.00	0.064	1.29	1.34	0.42	0.71	0.071
Middle SES	213 (30.4%)	2.29	2.36	0.75	1.00		1.43	1.38	0.40	0.57	
High SES	264 (37.7%)	2.29	2.27	0.74	1.14		1.43	1.42	0.39	0.57	
**Situation of siblings**											
Singletons	547 (78.03%)	2.29	2.34	0.74	1.07	0.514	1.43	1.40	0.40	0.57	0.157
Children with siblings	154 (21.97%)	2.36	2.38	0.77	1.21		1.29	1.34	0.42	0.71	
**C** **hildren** **’s** **gender**											
Boys	339 (48.36%)	2.36	2.39	0.76	1.14	0.116	1.29	1.33	0.40	0.71	0.000
Girls	362 (51.64%)	2.29	2.31	0.73	1.07		1.43	1.44	0.40	0.61	
**C** **hildren** **’s** **age**											
<3 years	46 (6.56%)	2.07	2.12	0.78	1.20	0.055	1.14	1.13	0.48	0.61	0.000
3–6 years	495 (70.61%)	2.29	2.35	0.74	1.07		1.43	1.40	0.39	0.57	
>6 years	160 (22.82%)	2.43	2.42	0.72	1.13		1.43	1.42	0.40	0.57	

**Table 2 behavsci-14-00436-t002:** Descriptive statistics and correlations of the main variables.

Variables	*M*	*SD*	1	2	3	4	5	6
1. Age	5.20	1.87	-					
2. Gender	0.52	0.50	-	-				
3. Electronic media use	2.35	0.74	0.11 **	−0.06	-			
4. Parent–child closeness	3.73	0.65	0.06	0.04	−0.15 ***	-		
5. Emotion regulation	3.32	0.47	0.05	0.07	−0.21 ***	0.64 ***	-	
6. Prosocial behavior	1.39	0.41	0.11 **	0.13 ***	−0.16 ***	0.59 ***	0.62 ***	-

Notes. Boy = 0, Girl = 1, ** *p* < 0.01, *** *p* < 0.001.

**Table 3 behavsci-14-00436-t003:** Bootstrap analysis of the mediation effect significance in the proposed model.

	Effect Value	95% Confidence Interval	
		Boot LLCI	Boot ULCI
Electronic media us→Parent–child closeness→Prosocial behavior	−0.0282	−0.06	−0.01
Electronic media us→Emotion regulation→Prosocial behavior	−0.1249	−0.19	−0.07
Electronic media us→Parent–child closeness→Emotion regulation→Prosocial behavior	−0.0744	−0.12	−0.03
Total indirect effect	−0.2275	−0.30	−0.15

Notes. When parent−child closeness and emotion regulation are included as mediators in the model, the direct effect of electronic media use on prosocial behavior is not significant (β = 0.04, *p* = 0.290, and 95% CI [−0.03, 0.12]). However, without accounting for these mediating variables, the direct effect is significant (β = −0.19, *p* < 0.001, and 95% CI [−0.27, −0.10]), suggesting full mediation within the tested model.

## Data Availability

The data from this study can be obtained by contacting the corresponding author.

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
