# Peer review of "Interplay between Children’s Electronic Media Use and Prosocial Behavior: The Chain Mediating Role of Parent–Child Closeness and Emotion Regulation"

_behavsci, 2024, doi:10.3390/bs14060436_

Round 1
Reviewer 1 Report
Comments and Suggestions for Authors
The authors addressed a very current problem and their findings largely confirm the cited reports from other studies.
The research results indicate an urgent need for psychologists - and decision-makers in the area of social policy, including educational and pro-family - to develop a concept of activities aimed at supporting the educational competences of parents/guardians of children, so as to avoid distortions of the socialization process of children, their emotional development and weakening of bonds in family. Failure to do so may result in negative long-term consequences for the development of individuals and society.
Generally, based on the research results, two paths of individual and social development of the youngest generation can be predicted - largely dependent on the family environment.
In general, the results confirm the importance of the parent-child relationship, the atmosphere of the family home and the socio-emotional maturity of parents/guardians for the child's development.
Reviewer 2 Report
Comments and Suggestions for Authors
The paper INTERPLAY BETWEEN CHILDREN'S ELECTRONIC MEDIA USE AND PROSOCIAL BEHAVIOR: THE CHAIN MEDIATING ROLE OF PARENT–CHILD CLOSENESS AND EMOTION REGULATION concerns very up-to -date and socially important topic. Critical discussion on technology and media profits and threats seems to be crucial question in mental health promotion. The aim of the study was to explore intercorrelation between children's electronic media use and prosocial behaviour mediated by parent–child closeness and emotion regulation. The research aimed at verifying 4 hypotheses. Participants comprised 701 parents who completed a structured questionnaire , among whom there were 73% mothers from northern regions of China. The mean age of children was 5.20 years.
In research material analyses chain mediation model was tested. As statistical tools PSS software (version 23.0) in variables interrelationships was employed and 306 AMOS (version 26.0) for structural equation modeling (SEM) was used. Obtained results support all the hypotheses .The results showed a significant mediated effect from children's electronic media use to prosocial behaviour via parent–child closeness (standardized effect = −0.0282, 95% CI [−0.06, −0.01]) and via emotion regulation (standardized effect = −0.1249, 95% CI [−0.19, −0.07]). Additionally the chain-mediated path- way – from electronic media use through parent–child closeness and emotion regulation to prosocial behavior – was also significant (standardized effect = −0.0744, 95% CI [−0.12, 380 −0.03]). Comprehensive strategies involving parents and educators in supporting children’s development in the digital age have been proposed as implications of important findings.
Strengths of the article
1. Very important and actual topic of the study . Mental health threats appearing in childhood ought to be intensively explored.
2. Big sample N= 701 (parents from the northern regions of China).
3. Impressive methodology : Hypothetical chain mediation model with aim to explore the complex dynamics of electronic media role in children's social and emotional development.
4.Well considered theoretical background: Individual-Environment Interaction Theory and Bi-Directionality of Parent–Child Relationships Theory.
5. Important variables treated as mediators in the research design, namely parent–child closeness and emotion regulation
Weaknesses of the article
1. Not balanced sample , 73 % mothers.
2. Type of media and content of media used by children wasn't described. Video games , Internet or TV programmes could be designed intentionally for children (Baby Einstein, · Sesame Street etc). This data ought to be controlled.
3. Important in-child factor crucial for prosocial development is temperament, what wasn’t controlled.
4. It will be interested to replicate the study with parents of older children . For example the comparison between group with preschool children, school children and adolescents.The highest value will have the longitudinal research design with observation of long-term dynamic between media use and prosocial development consequences.
The novelty of the research
Electronic media usage by children seems to be unavoidable in present time. Therefore deeper analyses of developmental consequences in children are important task for social sciences. Presented research outcomes illustrate the significance of the chain mediated effect, with a cascading influence: higher electronic media use in children is associated with reduced parent–child closeness and increased emotional challenges, adversely affecting prosocial behavior. Valuable suggestions for parents and educators were presented as conclusions: strong family ties and effective emotional management could be promoted via balanced approach to digital consumption ,outdoor activities, reading, and face-to-face social interactions. Information about connections between intensive media consumption and lower prosocial development in children should be disseminated for parents in preschool and school environment.
Reviewer 3 Report
Comments and Suggestions for Authors
The link between children's electronic media consumption and their prosocial behavior, particularly investigating the intermediary functions of parent-child closeness and emotion regulation has been investigated with comprehensive analyses. However, here are the major topics that need to be explained;
It should be clarified where the participants were selected from, how the survey was conducted (online surveys or paper-pen questionnaires, etc.), and where (hospital, school, etc.) it was administered and the timeframe during which data was collected.
The calculation method of the sample should be specified.
Family and child characteristics such as *child's attendance to daycare, *parents' level of education, *family type (nuclear, extended), *presence of mental health issues in parents, and the child's health status should be added to the study. If these variables are not included, the reasons for their omission should be explained in the limitations section.
Previous report has been reported that excessive media usage is associated with emotional lability (doi: 10.5546/aap.2021.eng.106). Excessive media usage and decreased parent-child closeness may be a result of parental neglect behavior (doi: 10.1111/jpc.14821). These aspects should be addressed in the introduction and discussion sections of the article.
Round 2
Reviewer 3 Report
Comments and Suggestions for Authors
The initial evaluation partially addresses concerns; however, further corrections are necessary. The statement, "This analysis showed that 267 our sample of 701 participants provides a power value of 0.84, surpassing the commonly accepted threshold of 0.80," requires a reference to substantiate its claim.
Additionally, the number of surveys administered and the reasons for exclusion from the study should be clearly stated.
Furthermore, it is essential to present the general characteristics of cases, sociodemographic features, and the results of independent and dependent variables according to the distribution characteristics in the form of a table, with values expressed as n, %, Mean/Median, Standard Deviation/Interquartile Range (IQR). Reference to the table should be made in the text, providing a brief summary of its contents.
